# Loss of Proximal Tubular Sirtuin 6 Aggravates Unilateral Ureteral Obstruction-Induced Tubulointerstitial Inflammation and Fibrosis by Regulation of β-Catenin Acetylation

**DOI:** 10.3390/cells11091477

**Published:** 2022-04-27

**Authors:** Jixiu Jin, Wenjia Li, Tian Wang, Byung-Hyun Park, Sung Kwang Park, Kyung Pyo Kang

**Affiliations:** 1Department of Internal Medicine, Research Institute of Clinical Medicine, Jeonbuk National University Medical School, Jeonju 54907, Korea; gilsoo1215@gmail.com (J.J.); liwenjia1214@gmail.com (W.L.); tianw0226@gmail.com (T.W.); parksk@jbnu.ac.kr (S.K.P.); 2Department of Biochemistry and Molecular Biology, Jeonbuk National University Medical School, Jeonju 54907, Korea; bhpark@jbnu.ac.kr; 3Biomedical Research Institute, Jeonbuk National University Hospital, Jeonju 54907, Korea

**Keywords:** kidney fibrosis, Sirt6, β-catenin, acetylation, TGF-β1/Smad signaling pathway

## Abstract

Renal fibrosis is a significant pathologic change associated with progressive kidney disease. Sirt6 is an NAD^+^-dependent deacetylase and mono-ADP ribosyltransferase known to play diverse roles in the processes attendant to aging, metabolism, and carcinogenesis. However, the role of proximal tubule-specific Sirt6 in renal fibrosis remains elusive. This study investigates the effect of proximal tubule-specific Sirt6 knockdown on unilateral ureteral obstruction (UUO)-induced renal tubulointerstitial inflammation and fibrosis. Renal fibrosis in wild type and *PT*-Sirt6KO (*Sirt6*^flox/flox^; *Ggt1-Cre^+^*) mice was induced by UUO surgery. After seven days, histologic examination and Western blot analysis were performed to examine extracellular matrix (ECM) protein expression. We evaluated inflammatory cytokine and cell adhesion molecule expression after ureteral obstruction. The therapeutic effect of Sirt6 activator MDL-800 on UUO-induced tubulointerstitial inflammation and fibrosis was assessed. The loss of Sirt6 in the proximal tubules aggravated UUO-induced tubular injury, ECM deposition, F4/80 positive macrophage infiltration, and proinflammatory cytokine and chemokine expression. Sirt6 activator MDL-800 mitigated UUO-induced renal tubulointerstitial inflammation and fibrosis. In an in vitro experiment, MDL-800 decreases the transforming growth factor (TGF)-β1-induced activation of myofibroblast and ECM production by regulating Sirt6-dependent β-catenin acetylation and the TGF-β1/Smad signaling pathway. In conclusion, proximal tubule Sirt6 may play an essential role in UUO-induced tubulointerstitial inflammation and fibrosis by regulating Sirt6-dependent β-catenin acetylation and ECM protein promoter transcription.

## 1. Introduction

Chronic kidney disease (CKD) affects more than 10~15% of the world’s population and is steadily growing as a global health concern [1,2]. The causes and mechanisms that lead to progressive kidney failure and associated systemic complications, such as cardiovascular disease, remain poorly understood, resulting in few targeted therapies other than renal replacement therapy [1,2,3]. Despite a promising preclinical study, holistic therapeutic interventions targeting human cytokines, transcription factors, signaling pathways, and epigenetic modulators have seen disappointing [3,4,5,6]. The situation is compounded by a paucity of additional therapeutic modalities [3]. Research into novel therapeutic targets might prove critical to saving lives and understanding new pathophysiologies associated with CKD.

Tubulointerstitial inflammation and fibrosis are significant pathologic changes associated with progressive kidney disease that result in end-stage renal disease. The complicated networks of the signaling pathways are involved in the renal fibrosis process, including Wnt/β-catenin signaling [7]. After an injury to a kidney, the transient activation of Wnt/β-catenin signaling may assist with the reparative process. However, sustained activation of Wnt/β-catenin signaling promotes fibrosis in the damaged kidney [8]. In renal fibrosis, an upregulated Wnt/β-catenin pathway leads to the proliferation and differentiation of fibroblasts into myofibroblasts, as it increases extracellular matrix (ECM) protein production and deposition in tubulointerstitial areas [9,10]. In addition, the epigenetic regulation of β-catenin has the potential to affect β-catenin target gene expression and trigger β-catenin signaling to promote fibrosis-related gene transcription and renal tubulointerstitial fibrosis [7]. 

In mammals, sirtuins (Sirt1~7) are an evolutionarily conserved family of enzymes with NAD^+^-dependent deacetylase, ADP-ribosyl transferase, and demalonylase/desuccinylase activity [11,12,13]. Sirtuins modify histone or non-histone proteins, thereby regulating various physiologic processes such as cell proliferation, DNA damage and repair, metabolism, and antioxidant activity [14]. Sirt1 activation by either resveratrol or Sirt1 agonist (SRT3025) decreases unilateral ureteral obstruction (UUO), 5/6 nephrectomy-induced renal inflammation, and ECM accumulation by Smad3 acetylation [15,16]. Sirt3 decreases aging-associated tissue fibrosis by activating glycogen synthase kinase (GSK)3β [17]. The activation of Sirt3 via honokiol mitigates UUO-induced renal fibrosis by regulating mitochondrial dynamics and transforming the growth factor (TGF)-β1/Smad signaling pathway [18]. Sirt6 also performs an essential role in maintaining the glomerular homeostasis in proteinuric kidney disease. Podocyte-specific Sirt6 deficiency exacerbates diabetic nephropathy and adriamycin-induced nephropathy by inhibiting Notch1 and Notch4 transcription through regulation of histone H3K9 acetylation [19]. Tian et al. have shown that Sirt6 is a critical modulator of cardiac fibroblast differentiation, as it regulates the nuclear factor-κB (NF-κB) signaling pathway [20]. However, understanding the role of proximal tubule-specific Sirt6 in renal fibrosis remains elusive.

In this study, we generated proximal tubule-specific Sirt6 knockout mice and evaluated the role of proximal tubular Sirt6 on UUO-induced tubulointerstitial inflammation and fibrosis. We also evaluated the therapeutic effect of the Sirt6 activator MDL-800 on UUO-induced tubulointerstitial inflammation and fibrosis. Our results suggest that the loss of Sirt6 in proximal tubules aggravates UUO-induced renal fibrosis and that Sirt6 activator MDL-800 mitigates UUO-induced renal tubulointerstitial inflammation and fibrosis by regulating Sirt6-dependent β-catenin acetylation and ECM protein transcription.

## 2. Materials and Methods

### 2.1. Animal Experiment

The Institutional Animal Care and Use Committee of Jeonbuk National University Hospital reviewed and approved the experimental animal protocol (cuh-IACUC-2018-3-1). *Sirt6*^flox/flox^ (*Sirt6*^tm1.1Cxd^/J, Stock No:017334) and γ-glutamyl transferase Cre mice (*Tg-(Ggt1-Cre)M3Egn/J*) were obtained from The Jackson Laboratory (Bar Harbor, ME, USA). Iwano et al. reported that Cre transcripts are only in the kidney, and after postpartum day 14, of γ-glutamyl transferase Cre transgenic mice express [21]. *Sirt6*^flox/flox^ and homozygous γ-glutamyl transferase Cre mice were crossed to obtain proximal tubule-specific Sirt6 KO mice (*PT*-Sirt6KO, *Sirt6*^flox/flox^; *Ggt1-Cre^+^**)* and littermate wild-type mice (WT, *Sirt6*^flox/flox^; *Ggt1-Cre^-^*). We made polymerase chain reactions (PCRs) with genomic DNA from mouse tails to confirm the genotype of each mouse (Appendix A). *PT*-Sirt6KO and WT mice were maintained in a room under conditions of controlled humidity (40~60%), temperature (23 ± 1 °C), and lighting (12 h-light and dark cycle), with food and water provided ad libitum. For the animal experiment, we divided the mice into four groups: sham operation in WT mice (WT Con), UUO operation in WT mice (WT UUO), sham operation in *PT*-Sirt6KO mice (*PT*-Sirt6 Con), and UUO operation in *PT*-Sirt6KO mice (*PT*-Sirt6 UUO) (n = 15/group). 

To evaluate the therapeutic effect of Sirt6, C57/BL6 mice (7–8 weeks old; weight 20–25 g, Orient Bio, Inc.; Seoul, Korea) were treated with a Sirt6 activator, MDL-800 (Sigma-Aldrich; Merck KGaA, Darmstadt, Germany) one day after UUO surgery. We divided the mice into four groups for this experiment: sham + vehicle (Veh Con), sham + MDL-800 (MDL Con), UUO + vehicle (Veh UUO), and UUO + MDL-800 (MDL UUO) group (n = 15/group). MDL-800 was dissolved in a mixture of 5% dimethyl sulfoxide (DMSO), 30% polyethylene glycol (PEG)-400 (Sigma-Aldrich), and 65% phosphate-buffered saline (PBS) [22]. We used a mixture of 5% DMSO, 30% PEG-400, and 65% PBS without MDL-800 as a vehicle (Veh). We administered MDL-800 (10 mg/kg) or Veh by daily intraperitoneal injections that began one day after UUO and ended 7 d after surgery. 

We performed a UUO surgery using previously described methods [18,23,24]. In brief, mice were anesthetized via intraperitoneal injection with ketamine (100 mg/kg, Yuhan, Seoul, Korea) and xylazine (10 mg/kg, Bayer Korea, Seoul, Korea). After checking the pedal reflex, mice were placed on a temperature-controlled operating table with body temperature maintained at 37 °C. The right proximal ureter was dissected and ligated at two separated points using 3–0 black silk after a midline abdomen incision. The sham operation was made using the same method without ligation of the ureter. Seven days after UUO surgery, the obstructed and contralateral kidneys were harvested, prepared for histologic examination, and stored separately at −80 °C for Western blot analysis and real-time PCR. 

### 2.2. Renal Histology

UUO and contralateral kidneys were fixed in 4% paraformaldehyde and embedded in paraffin. The paraffin block was prepared into 5 μm sections, each of which was stained with Hematoxylin and Eosin (H&E), and Masson’s trichrome (MTC) [24]. We then performed immunohistochemical consistent with previously described methods [18,24]. Slide sections were deparaffinized using xylene, rehydrated via graded washes of ethanol in water, and rinsed in pure water. We next performed a heat-induced antigen retrieval process and treated blocking buffer. The tissue slides were incubated overnight at 4 °C with rabbit anti-mouse monocyte chemoattractant protein-1 (MCP-1; 70R50662; dilution 1:50; Fitzgerald Industries International, Acton, MA, USA) and hamster anti-mouse intercellular adhesion molecule-1 (ICAM-1; 553249; dilution 1:100; BD Biosciences, San Jose, CA, USA). We treated the kidney sections with DAKO Chromogen (DAKO Cytomation, Glostrup, Denmark) to visualize immune complexes and then counterstained them with hematoxylin (Sigma-Aldrich Co., Burlington, MA, USA). 

After preparing freshly frozen renal tissues, the slides were fixed with 4% paraformaldehyde, permeabilized with Triton X-100 (1%), and then incubated with a blocking buffer for immunofluorescence staining [24]. The tissue samples were incubated with rabbit anti-Sirt6 (12486; 1:50, Cell Signaling Technology, Inc. Danvers, MA, USA), FITC-labeled *Lotus tetragonolobus* lectin (Vector Laboratories, Burlingame, CA, USA), a mouse anti-α-smooth muscle actin (α-SMA; A2547; 1:100; Sigma-Aldrich), anti-fibroblast specific protein-1 (FSP-1; ab27957; 1:100; Abcam, Cambridge, UK), and F4/80 (14-4801-82; 1:200; eBioscience, San Diego, CA, USA), after which they were exposed to Cy3-labeled secondary antibody (Chemicon, Temecula, CA, USA). For nuclear staining, the kidney sections were incubated with 300 nM of 4′,6-diamidino-2-phenylindole solution for 3 min (DAPI; Molecular Probes; Thermo Fisher Scientific Inc. Waltham, MA, USA). For histologic and morphometric analyses, two observers who were unaware of the origins of the samples, respectively, used a Carl Zeiss Z1 microscope or Carl Zeiss LSM 880 super-resolution confocal laser scanning microscope (Carl Zeiss, Göttingen, Germany) to evaluate each slide. A tubular injury score was counted into six levels based on tubular dilation, epithelial desquamation, and loss of brush border as assessed at ten randomly chosen, non-overlapping fields at a magnification of 200×: 0, none; 0.5, <10%; 1, 10–25%; 2, 25–50%; 3, 50–75%; and 4, >75% [23,24]. The fibrotic area, ICAM-1, and MCP-1 positive areas were also measured in ten randomly chosen non-overlapping fields at a magnification of 200×. The area fraction of α-SMA was measured at a magnification of 400×. The numbers of FSP-1(+) fibroblasts and F4/80 (+) macrophages were counted at a magnification of 400×. All images were analyzed using ImageJ software (http://rsb.info.nih.gov/ij, accessed on 12 April 2022).

### 2.3. Picro Sirius Red Stain

The kidney sections were stained with Picro Sirius red to evaluate collagen deposition after ureteral obstruction [18,25]. After deparaffinization, the tissue sections were hydrated and stained with 0.1% Picro Sirius red solution (Sigma-Aldrich Co.) for 1 h, then washed in acidic water and dehydrated and mounted. Under 100× magnification, the Picro Sirius red-positive areas were measured in ten randomly chosen non-overlapping fields using ImageJ software.

### 2.4. Western Blotting and Immunoprecipitation

Western blot analysis was performed consistent with previously described methods [24,26]. Kidney tissues and cell lysates were separated by 7~12% SDS-PAGE. After electrophoresis, the samples were transferred into PVDF membranes (Bio-rad, Hercules, CA, USA) and blocked with 5% skim milk (Bio-rad). We probed each blot with Sirt6 primary antibodies (12486; rabbit; 1:1000, Cell Signaling Technology Inc.), α-SMA (A2547; mouse; 1:1000; Sigma-Aldrich Co.), type I collagen (1310-01; goat; 1:1000; Southern Biotech, Birmingham, AL, USA), CTGF (sc-365970; mouse; 1:1000; Santa Cruz Biotechnology, Inc., Dallas, TX, USA), fibronectin (sc-8422; mouse; 1:1000; Santa Cruz Biotechnology, Inc.), acetyl-β-catenin (Lys49) (9030; rabbit; 1:1000, Cell Signaling Technology Inc.), active-β-catenin (8814; rabbit; 1:1000, Cell Signaling Technology Inc.), phospho-Smad2 (3101; rabbit; 1:1000; Cell Signaling Technology Inc.), phospho-Smad3 (9520; rabbit; 1:1000; Cell Signaling Technology Inc.), Smad2/3 (07-408; rabbit; 1:1000; EMD Millipore, Billerica, MA, USA), β-actin (sc-47778; mouse; 1:2000; Santa Cruz Biotechnology, Inc.), and glyceraldehyde 3-phosphate dehydrogenase (GAPDH; AP0063; rabbit; 1:2000; Bioworld Technology, Inc., Danvers, MA, USA). Both β-actin and GAPDH were used as an internal control. 

For immunoprecipitation, 1 mL of whole-cell lysate with 2 μg of either control IgG or primary antibody was incubated first for 24 h at 4 °C and then subsequently with protein A/G PLUS-agarose (Santa Cruz Biotechnology, Inc.) for 2 h at 4 °C. After collecting and washing the pellet, cell lysates were separated by 10% SDS-PAGE and transferred to the PVDF membrane. The blot was probed with primary antibodies. All signals were visualized by a chemiluminescent detection kit (Amersham Pharmacia Biotech, London, UK) and analyzed by a densitometric scanner (ImageQuant LAS 4000 Mini, GE Healthcare Life Sciences, Piscataway Township, NJ, USA).

### 2.5. RNA Isolation and Quantitative PCR Analyses

RNA was isolated, and quantitative real-time reverse transcription polymerase chain reaction (qRT-PCR) was performed consistent with methods described previously [27,28]. Briefly, total RNA was isolated from frozen kidney homogenate using TRIzol reagent (Invitrogen, Carlsbad, CA, USA). A Transcriptor First Stand cDNA Synthesis Kit (Roche Diagnostic, Mannheim, Germany) was used in accordance with the manufacturer’s instructions to synthesize cDNA from the total RNA. qRT-PCR of mouse *Tnf-α, Il-1β, Icam-1, Mcp-1, Tgf-β1, Fibronectin, Col1A1, and Col3A1 mRNA* was performed in a Rotor-Gene Q (Qiagen, Germantown, MD, USA). The specific primers for each gene (Table 1) were designed using Primer-BLAST (NCBI, www.ncbi.nlm.nih.gov/tools/primer-blast/, accessed on 12 April 2022). A 10-fold dilution of each cDNA transcript was amplified in a 10-μL volume, using SYBR^®^ Green PCR Master Mix (Applied Biosystems, Inc., Carlsbad, CA, USA), resulting in 200 nM final concentration of each primer. The PCR program was 10 min at 95 °C, followed by 40 cycles of 95 °C for 10 s and 60 °C for 30 s. To confirm the use of equal amounts of RNA in each reaction, we examined all samples in parallel for GAPDH mRNA expression.

### 2.6. Cell Culture Experiment

In vitro experiments were performed using an immortalized human proximal tubule epithelial cell line (HK-2 cells, American Type Culture Collection, Manassas, VA, USA). We cultured HK-2 cells in Dulbecco’s modified Eagle’s medium/F12 supplemented with 10% (vol/vol) heat-inactivated fetal bovine serum and antibiotics (100 U/mL penicillin G and 100 μg/mL streptomycin; Gibco, Waltham, MA, USA) at 37 °C with 5% CO_2_ in 95% air. To investigate the effect of MDL-800 on TGF-β1-induced ECM expression and the activation of the TGF-β1/Smad signaling pathway, we incubated sub-confluent HK-2 cells with MDL-800 (10, 25, or 50 μM) for 30 min and then stimulated them with TGF-β1 (10 ng/mL, Sigma-Aldrich Co.) for 24 h.

### 2.7. Cell Proliferation Assay

After 24 h treatment with MDL-800 (10, 25, or 50 μM) and TGF-β1 (10 ng/mL), the proliferation of HK-2 cells was determined by colorimetric assay (Cell Proliferation Kit II, Roche Diagnostics, Mannheim, Germany), performed consistent with the manufacturer’s protocol. All experimental values were determined from triplicate wells.

### 2.8. Chromatin Immunoprecipitation (ChIP) Assay

ChIP assay was performed using an EZ ChIP chromatin immunoprecipitation kit (Millipore, Temecula, CA, USA) consistent with the manufacturer’s instructions. In brief, HK2 cells were incubated with MDL-800 (50 μM) for 30 min and then stimulated with TGF-β1 (10 ng/mL) for 24 h. To cross-link acetyl-β-catenin to DNA, cells were treated with 1% formaldehyde for 10 min at 37 °C. After cells were washed and harvested, genomic DNA fragments were obtained by sonicating cell lysates. The cross-linked acetyl-β-catenin-DNA complexes were immunoprecipitated using an antibody against acetyl-β-catenin and normal rabbit IgG as negative controls. After precipitation with salmon sperm DNA/protein A agarose slurry, the acetyl-β-catenin-DNA complexes were eluted and reversed by heating at 65 °C for 4 h. The DNA was recovered by phenol/chloroform extraction. The inputs consist of 5% chromatin before immunoprecipitation. The ChIP enriched DNA and input DNA were analyzed by real-time PCR using the following primers: *Fn1* promoter forward 5′- TCCTTCCCCCAGAATCAATGAA-3′, reverse 5′-GGGAAGCCGAGTGTTTCTTCC-3′; *MMP-7* promoter forward 5′-CAGCACGGTGAGTCGCATA-3′, reverse 5′- TTTCCACATTCGAGGCTGAG-3′; and *Snail* promoter forward 5′- ACTATGCCCACCGACCCT-3′, reverse 5′-CCAGACCTTTCCCACCTT-3′. The relative enrichment of promoter DNA was normalized to the input. 

### 2.9. Statistical Analyses

The data are expressed as the mean ± standard deviation (SD). We used the Shapiro–Wilk test to confirm whether the data set was normally distributed. A one-way analysis of variance (ANOVA) was used for normally distributed data to evaluate differences within groups, followed by an individual comparison between groups with the Tukey post hoc test. For non-parametric data, the Kruskal–Wallis one-way ANOVA on ranks was used, followed by all multiple pairwise comparisons with the Dunn’s method. We used SigmaPlot ver. 14.0 software (Systat Software Inc.; Berkshire, UK) for statistical analysis and graph processing to present the data. *p* < 0.05 was considered statistically significant.

## 3. Results

### 3.1. Loss of Proximal Tubule Sirt6 Exacerbates UUO-Induced Tubular Injury and Fibrosis

In a previous report, pharmacologic inhibition of Sirt6 was observed to aggravate UUO-induced renal tubular injury, fibrosis, and inflammation [7]. To understand the role of proximal tubule Sirt6 in UUO-induced renal tubulointerstitial inflammation and fibrosis, we generated proximal tubule-specific Sirt6 knockout mice by the crossing of *Sirt6*^flox/flox^ and γ-glutamyl transferase Cre mice (*Tg-(Ggt1-Cre)M3Egn/J*) (Appendix A). To confirm proximal tubule-specific Sirt6 knockout (*PT*-Sirt6KO), we performed a double immunofluorescence stain for Sirt6 and proximal tubular maker *Lotus tetragonolobus* lectin in sham and UUO kidneys. In the sham-operated *PT*-Sirt6KO mice, Sirt6 expression was significantly lower than that of wild-type (WT) mice in the proximal tubule cells. Sirt6 expression was elevated in the proximal tubule of the WT UUO kidney when compared to the sham-operated WT mice. However, decreased Sirt6 expression was observed in the proximal tubules of the *PT*-Sirt6KO UUO kidney compared to the WT UUO kidney (Figure 1A). We confirmed Sirt6 expression after UUO surgery in the WT and *PT*-Sirt6KO mice by Western blot analysis. In the sham-operated *PT*-Sirt6KO kidney, Sirt6 expression was significantly lower than that in the WT kidney, while Sirt6 expression had significantly increased post-surgery compared to the sham-operated WT kidney. However, in the *PT*-Sirt6KO UUO kidney, Sirt6 expression was lower than in the WT UUO kidney.

Histologically, neither WT nor *PT*-Sirt6KO mice show tubular abnormalities in the sham-operated kidneys. However, the *PT*-Sirt6KO UUO kidney showed more tubular dilatation, loss of brush border, and inflammatory cells infiltrations than the WT UUO kidney after 7 d of ureteral obstruction. We evaluated UUO-induced renal fibrosis using Masson’s trichrome stain (MTC), which revealed a significant increase in fibrotic area in the *PT*-Sirt6KO UUO kidney but not in the WT UUO kidney (Figure 2). These data suggest that proximal tubule-specific Sirt6 plays an essential role in UUO-induced tubular injury, inflammation, and fibrosis.

### 3.2. Loss of Proximal Tubule Sirt6 Increases Myofibroblast Activation and Extracellular Matrix Deposition

Myofibroblast activation and ECM deposition are important pathophysiologic mechanisms associated with renal fibrosis [18,23,24,25]. Therefore, through immunofluorescence staining for α-SMA and FSP-1 expression after 7 d of ureteral obstruction, we investigated whether proximal tubule Sirt6 has a role in UUO-induced myofibroblast activation. The UUO-induced increase in α-SMA and FSP-1 expression was significantly higher in the *PT*-Sirt6KO UUO kidney than in the WT UUO kidney (Figure 3A). The Western blot analysis revealed that UUO-induced α-SMA expression was higher in the *PT*-Sirt6KO kidney than in the WT kidney (Figure 3B). 

We also evaluated ECM deposition by Picro Sirius red stain. Seven days after ureteral obstruction, Picro Sirius red-positive area had significantly increased in the *PT*-Sirt6KO UUO kidneys but not in the WT UUO kidneys (Figure 4A). In addition, the expression of type I collagen was significantly increased in the *PT*-Sirt6KO UUO kidney but not in the WT UUO kidney. Connective tissue growth factor (CTGF) had increased at 7d after ureteral obstruction in both WT and *PT*-Sirt6KO kidneys, though its expression was significantly higher in the *PT*-Sirt6KO UUO kidneys than in the WT UUO kidneys (Figure 4B). These data suggest that proximal tubule Sirt6 has an essential role in UUO-induced myofibroblast activation and ECM deposition. 

### 3.3. Loss of Proximal Tubule Sirt6 Increases Inflammatory Cells Infiltration, as Well as Proinflammatory Cytokine and Chemokine Expression

Triggered by significant tissue damage, an inflammatory response induces both tissue regeneration and the development of fibrosis [18,29]. Therefore, we evaluated whether proximal tubule Sirt6 has a role in UUO-induced renal tubulointerstitial inflammation. The infiltration of F4/80 (+) macrophages was observed to have increased in the *PT*-Sirt6KO UUO kidneys but not in the WT UUO kidneys (Figure 5A). 

We evaluated the mRNA levels of proinflammatory cytokines and chemokines by qRT-PCR. *Tnf-**α, Il-1**β, Icam-1*, and *Mcp-1* mRNA expression were significantly increased in the WT and the *PT*-Sirt6 UUO kidneys but not in the sham kidneys. However, the mRNA levels of proinflammatory cytokines and chemokines were more elevated in the *PT*-Sirt6KO UUO kidneys (Figure 5B). We also evaluated MCP-1 and ICAM-1 protein expression seven days after UUO surgery by immunohistochemistry. The UUO-induced increase in MCP-1 expression was higher in the *PT*-Sirt6KO kidney than in the WT kidney. MCP-1 expression was most noticeable in the tubules in the WT and *PT*-Sirt6KO UUO kidneys. UUO induced a greater increase in ICAM-1 expression in the *PT*-Sirt6KO kidney than in the WT kidney. ICAM-1 expression was most prominently in the tubulointerstitial area of the WT and *PT*-Sirt6KO UUO kidneys. These data suggest that proximal tubule Sirt6 performs an essential role in UUO-induced tubulointerstitial inflammation. 

### 3.4. Sirt6 Activator MDL-800 Exerts a Therapeutic Effect on UUO-Induced Renal Tubular Injury, Inflammation, and Fibrosis

We evaluated whether Sirt6 activator MDL-800 has a therapeutic effect on UUO-induced renal tubulointerstitial inflammation and fibrosis (Figure 6A). MDL-800 was administered one day after UUO surgery. In the UUO kidney, MDL-800 treatment decreased UUO-induced tubular dilatation, loss of brush border, and inflammatory cells infiltrations compared to levels below those observed in the Veh-treated group (Figure 6B). We also evaluated the mRNA levels of *Tgf-**β1, fibronectin, Col1A1,* and *Col3A1,* as well as the mRNA levels of proinflammatory cytokines and chemokines by qRT-PCR. The mRNA levels of *Tgf-**β1, fibronectin, Col1A1,* and *Col3A1* were significantly elevated in the Veh-treated UUO kidneys. MDL-800 treatment significantly repressed otherwise UUO-induced increases in mRNA levels of *Tgf-**β1*, *fibronectin, Col1A1,* and *Col3A1*. In the Veh-treated UUO kidney, mRNA levels of *Tnf-**α, Il-1**β, Icam-1,* and *Mcp-1* were significantly elevated. However, treatment with Sirt6 activator MDL-800 inhibited the UUO-induced increase in mRNA levels of proinflammatory cytokines and chemokines (Figure 6C).

We evaluated the effect of MDL-800 on UUO-induced myofibroblast activation, ECM deposition, and tubulointerstitial inflammation. MDL-800 treatment suppressed the UUO-induced increase in FSP-1 (+) and F4/80 (+) cell infiltrations (Figure 7A,C). In the Picro Sirius red stain, collagen depositions in the UUO kidneys were observed to have been reduced by treatment with MDL-800 (Figure 7B). MDL-800 treatment repressed the UUO-induced increase in MCP-1 and ICAM-1 expression compared to levels below those observed in the Veh-treated UUO kidney (Figure 7D). These data suggest that Sirt6 activation may have a protective effect on UUO-induced tubular injury, fibrosis, and inflammation. 

### 3.5. Sirt6 Activator MDL-800 Decreases TGF-β1-Induced Extracellular Matrix Protein Expression by Regulation of the β-Catenin Acetylation and TGFβ1-Smad Signaling Pathway in Human Proximal Tubule Cells

To understand the protective mechanism of MDL-800 on UUO-induced renal fibrosis, we evaluated TGF-β1-induced renal proximal tubular cell proliferation in vitro using the HK2 human proximal tubule cell line. Treatment with TGF-β1 (10 ng/mL) increased renal proximal tubular cell proliferation approximately 1.9-fold compared to Veh-treated cells. Treatment with MDL-800 significantly mitigated the TGF-β1-induced increase in renal proximal tubular cell proliferation in a dose-dependent manner (Figure 8A). 

We evaluated whether MDL-800 could modulate TGF-β1-induced ECM protein expression in HK-2 cells. After 24 h of stimulation, TGF-β1 (10 ng/mL) was observed to have significantly increased α-SMA, type I collagen, fibronectin, and CTGF expression in HK2 cells. MDL-800 significantly suppressed the TGF-β1-induced increase in α-SMA, type I collagen, fibronectin, and CTGF expression in a dose-dependent manner (Figure 8B). To ascertain the effect of MDL-800 on the TGF-β1/Smad signaling pathway, we also evaluated the effect of TGF-β1-induced phosphorylation of Smad2 and Smad3. MDL-800 curbed the TGF-β1-induced increase in Smad2 and Smad3 phosphorylation in a dose-dependent manner (Figure 8C).

We then evaluated Sirt6 target protein acetylation and the relationship between Sirt6 and its target protein. After 24 h stimulation with TGF-β1 (10 ng/mL), acetylation of β-catenin had significantly increased at lysine 49. MDL-800 suppressed the TGF-β1-induced increase in acetyl-β-catenin expression in a dose-dependent manner. Notably, active-β-catenin also increased after 24 h stimulation with TGF-β1 (10 ng/mL), which was subsequently inhibited by MDL-800 treatment (Figure 9A). To address the relationship between Sirt6 and β-catenin, we performed an immunoprecipitation assay using Sirt6 and β-catenin antibodies. Over the course of 24 h of stimulation with TGF-β1, Sirt6 interacted with β-catenin. The amount of β-catenin present was lowered by the MDL-800 treatment (Figure 9B). We further evaluated whether Sirt6-dependent β-catenin acetylation was associated with β-catenin target genes, such as *Fn1*, *MMP-7*, and *snail*, promoter enrichment in HK2 cells by Chromatin immunoprecipitation (ChIP) assay. The number of promoters of β-catenin target genes enrichment had increased after treatment of TGF-β1 compared to Veh or MDL-800-treated HK2 cells. However, MDL-800 significantly inhibited the TGF-β1-induced increase in *Fn1, MMP7,* and *Snail* promoter enrichment (Figure 9C). These data suggest that Sirt6 activator MDL-800 modulates TGF-β1-induced ECM expression in renal proximal tubules cells via Sirt6-dependent β-catenin acetylation and β-catenin target gene promoter enrichment at the site of β-catenin acetylation.

## 4. Discussion

Organ fibrosis in response to injury is essentially a self-limiting repair process to overcome tissue damage. After renal injuries, damaged tubular epithelial cells and capillary endothelial cells generate a signal to recruit inflammatory cells and activate fibroblasts through a genetic/epigenetic reprogramming process [30]. Previously, we found that HDAC1 inhibitor valproic acid decreases TGF-β1-induced myofibroblast activation by promoting histone (H3) acetylation at lysine 9 and 14 and by regulating *Fn1* and *Col1**α1* promoter enrichment at acetyl-histone (H3) [25]. This study used proximal tubule-specific Sirt6 KO mice to investigate whether Sirt6 in the proximal tubule exerts a beneficial effect on UUO-induced tubulointerstitial inflammation and fibrosis (*PT*-Sirt6KO, *Sirt6*^flox/flox^; *Ggt1-Cre^+^*). The novel finding of this study is that loss of Sirt6 in the proximal tubule aggravates UUO-induced renal tubulointerstitial inflammation and fibrosis. Moreover, we were able to show that the Sirt6 activator MDL-800 exercises a therapeutic effect on UUO-induced tubulointerstitial inflammation and fibrosis by regulating Sirt6-dependent β-catenin acetylation and the TGF-β1/Smad signaling pathway. 

Several research reports have focused on the role of sirtuins in the maintenance of renal homeostasis [11]. In the kidney, Sirt1 is the most widely studied, as it is expressed throughout the glomerulus and tubules and maintains structural and functional integrity [11,31,32]. Sirt3 is another important mitochondrial sirtuin, responsible for maintaining mitochondrial dynamics, energy homeostasis, and antioxidant defenses in proximal and distal tubules [18,33]. Sirt6 has been reported to cellular processes as diverse as DNA repair and genomic stability, metabolism and aging, and carcinogenesis [34]. In addition, histone and non-histone proteins are known targets for Sirt6 deacetylase activity [7,19,35]. In the kidney injury model, Sirt6 has different roles in the kidney cell specifically. Podocyte or endothelial cell-specific Sirt6 knockout mice have been used to illustrate the detrimental effects of the regulation of histone H3K9 acetylation on diabetic nephropathy or adriamycin-induced podocyte injury [19] and angiotensin II-induced hypertension and its complications [35]. Maity et al. reported that Sirt6 deficiency transcriptionally upregulates the TGF-β signaling pathway by Smad3 acetylation and results in spontaneous hyperactivation of myofibroblasts and multiorgan fibrosis [36]. Conversely, Sirt6 overexpression inhibits TGF-β-induced myofibroblast differentiation by inhibiting Smad2 nuclear translocation and reducing NFκB dependent gene expression [37]. In the present study, we use γ-glutamyl transferase Cre mice, which express Cre transcripts, including proximal tubules, only in their cortical kidneys, and not in other tissues, (e.g., brain, liver, spleen, lung, muscle, or adrenal gland). Additionally, their bone marrow cells do not express γ-glutamyl transferase Cre [21]. In a previous mouse single-cell transcriptomics study, approximately 60% of the kidney cells were found to be proximal tubule cells [38,39]. Immunofluorescence stain and Western blot analysis of our *PT*-Sirt6KO mice revealed that they showed significantly less Sirt6 expression than WT mice. We also showed that the loss of Sirt6 in proximal tubule cells aggravates UUO-induced myofibroblast activation, extracellular matrix deposition, and F4/80 positive macrophage infiltration. The expression of proinflammatory cytokines and chemokines was also observed to have increased in the *PT*-Sirt6KO UUO kidneys. Our data suggest that proximal tubule Sirt6 has an essential role in the UUO-induced tubulointerstitial inflammation and fibrosis process. 

Fibrosis is a failed repair process that occurs after organ injury, characterized by excessive ECM deposition, organ dysfunction, and failure [40]. The activation of the Wnt/β-catenin signaling pathway plays a crucial role in promoting renal fibrosis by controlling the expression of downstream mediators, the activation and proliferation of myofibroblasts, and the secretion and deposition of ECM in tubulointerstitial areas [8]. TGF-β stimulates canonical Wnt signaling in a p38 dependently, and this interaction has a crucial role in the pathogenesis of fibrotic diseases such as systemic sclerosis, idiopathic pulmonary fibrosis, and liver cirrhosis [41]. In the CKD model, blocking TGF-β and β-catenin crosstalk in the proximal tubules aggravates tubular injury and tubulointerstitial fibrosis [42]. Cai et al. reported that Sirt6 deacetylase histone H3K56 results in decreased β-catenin target gene expression [7], suggesting that the Wnt/β-catenin signaling pathway may be a potential target with therapeutic potential for the treatment of renal fibrosis. In our in vivo model, Sirt6 activator MDL-800 showed therapeutic effects on UUO-induced renal tubulointerstitial inflammation and fibrosis. Interestingly, our data showed that Sirt6 directly interacts with β-catenin and regulates β-catenin acetylation in HK2 cells. Sirt6 activator MDL-800 regulates TGF-β1-induced increases of acetyl-β-catenin and β-catenin target gene promoter enrichment at the site of β-catenin acetylation. MDL-800 also decreases TGF-β1-induced increases in Smad2 and 3 phosphorylation.

One limitation of this study was that we did not use inducible Cre-LoxP systems. However, our γ-glutamyl transferase Cre mice express Cre transcripts after postpartum day 14, which might be the point of complete end of nephrogenesis [21]. In addition, our *PT*-Sirt6KO mice show grossly and histologically normal findings. If further studies are required to address the role of Sirt6 in kidney disease, we may use a kidney-specific inducible Cre mouse or CRISPR system.

## 5. Conclusions

In conclusion, the present study showed that the loss of proximal tubule Sirt6 increases UUO-induced myofibroblast activation, ECM deposition, and proinflammatory cytokine and chemokine expression. Treatment with Sirt6 activator MDL-800, however, effectively impedes UUO-induced renal tubulointerstitial inflammation and fibrosis. In short, proximal tubule Sirt6 is suggested to exercise an essential role in UUO-induced tubulointerstitial inflammation and fibrosis by regulating Sirt6-dependent β-catenin acetylation and ECM promoter enrichment at the site of β-catenin acetylation.

## Figures and Tables

**Figure 1 cells-11-01477-f001:**
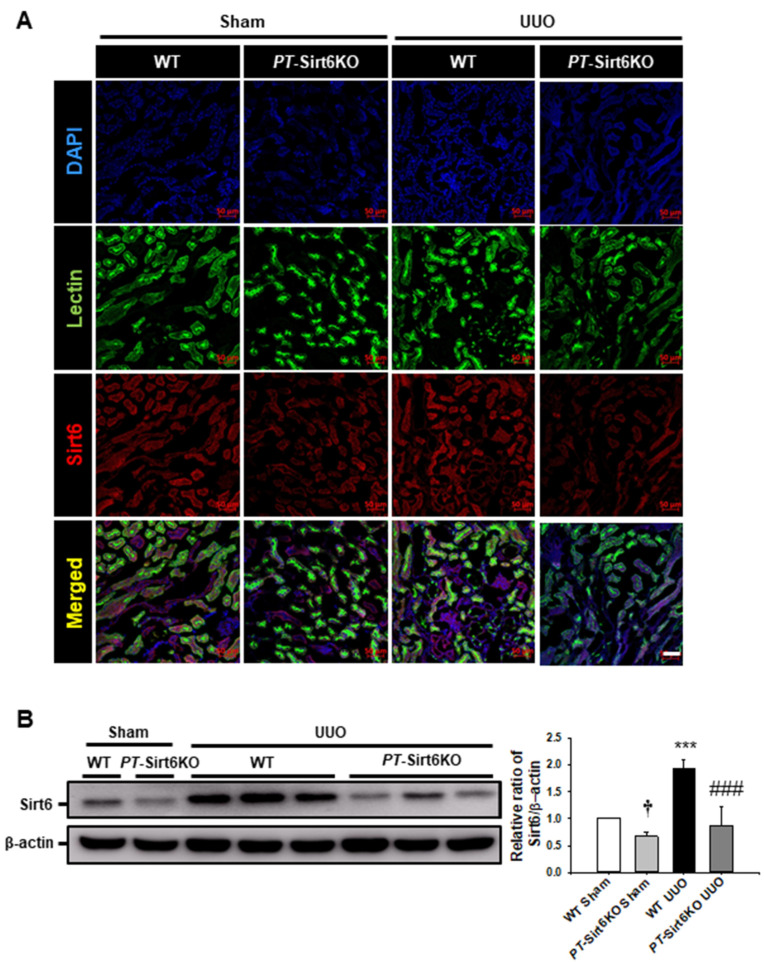
**Establishment of proximal tubule-specific Sirt6 knockout (*Sirt6*^flox/flox^; *Ggt1-Cre^+^*) mice** (**A**) Proximal tubule-specific loss of Sirt6 was confirmed by immunofluorescence staining of Sirt6 in proximal tubules. *Lotus tetragonolobus* lectin was used as a proximal tubule marker. Scale bar = 50 μm. (**B**) Representative Western blot analysis of Sirt6 in the kidneys from sham- and UUO-operated *Sirt6*^flox/flox^; *Ggt1-Cre^-^* (WT) and *Sirt6*^flox/flox^; *Ggt1-Cre^+^* (*PT*-Sirt6KO) mice. The bar graph shows the densitometric quantification presented as the relative ratio of each protein to β-actin. The relative ratio measured in the kidneys from sham-operated WT mice is arbitrarily presented as 1. Data are expressed as mean ± SD. †, *p* < 0.05 versus WT sham; ***, *p* < 0.001 versus WT sham or *PT*-Sirt6KO sham; ###, *p* < 0.001 versus WT UUO. Sham, sham-operated mice; UUO, unilateral ureteral obstruction; WT, wild-type; *PT*-Sirt6KO, proximal tubule-specific Sirt6 knockout; lectin, *Lotus tetragonolobus* lectin.

**Figure 2 cells-11-01477-f002:**
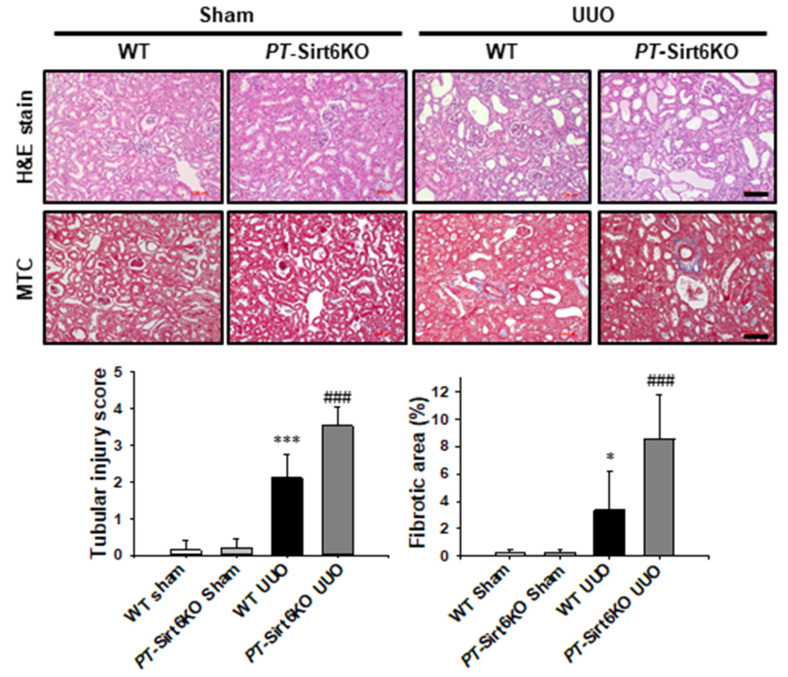
**Proximal tubule-specific loss of Sirt6 exacerbates UUO-induced tubular injury and fibrosis.** Representative sections of kidneys from sham- and UUO-operated *Sirt6*^flox/flox^; *Ggt1-Cre^-^* (WT) and *Sirt6*^flox/flox^; *Ggt1-Cre^+^* (*PT*-Sirt6KO) mice were stained with Hematoxylin and Eosin (H&E) and Masson’s trichrome (MTC). Scale bar = 100 μm. The bar graph shows semi-quantitative scoring of tubular injury by H&E, area fractions (%) of tubulointerstitial fibrosis, and degree of interstitial collagen deposition stained by MTC in the sham and UUO-operated kidneys at a magnification of 200×. Ten randomly chosen, non-overlapping fields were quantified (n = 15 per group). Data are expressed as mean ± SD. *, *p* < 0.05 and ***, *p* < 0.001 versus WT sham or *PT*-Sirt6KO sham; ###, *p* < 0.001 versus WT UUO. Sham, sham-operated mice; UUO, unilateral ureteral obstruction; WT, wild-type; *PT*-Sirt6KO, proximal tubule-specific Sirt6 knockout.

**Figure 3 cells-11-01477-f003:**
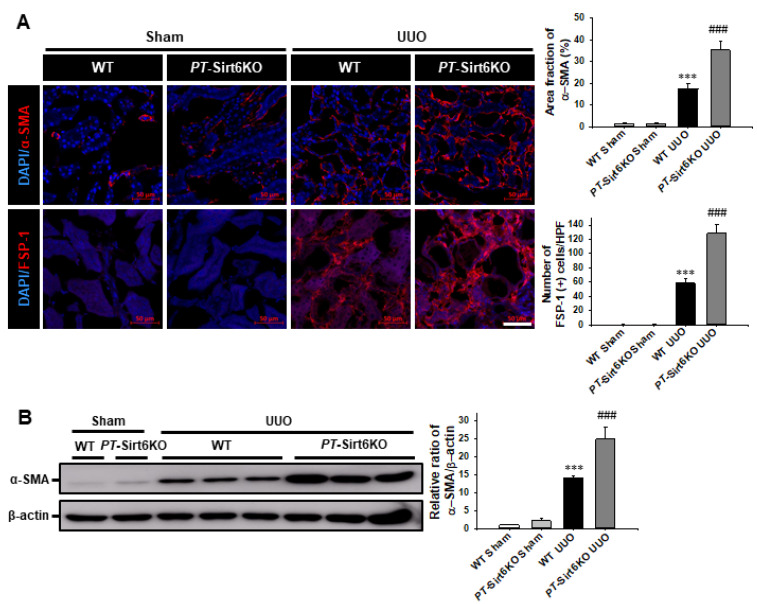
**Proximal tubule-specific loss of Sirt6 increases UUO-induced myofibroblast activation.** (**A**) Representative sections of kidneys from sham- and UUO-operated *Sirt6*^flox/flox^; *Ggt1-Cre^-^* (WT) and *Sirt6*^flox/flox^; *Ggt1-Cre^+^* (*PT*-Sirt6KO) mice were immunofluorescence stained with α-SMA and FSP-1 (red). The nucleus was stained by DAPI (blue). Scale bar = 50 μm. The bar graph shows area fractions of α-SMA (%) and the number of FSP-1 (+) cells in the sham- and UUO-operated kidneys from ten randomly chosen, non-overlapping fields at a magnification of 400× (n = 15 per group). (**B**) Representative Western blot analysis of α-SMA expression in the kidneys from sham- and UUO-operated *Sirt6*^flox/flox^; *Ggt1-Cre^-^* (WT) and *Sirt6*^flox/flox^; *Ggt1-Cre^+^* (*PT*-Sirt6KO) mice. The bar graph shows the densitometric quantification presented as the relative ratio of each protein to β-actin. The relative ratio measured in the kidneys from sham-operated WT mice is arbitrarily presented as 1. Data are expressed as mean ± SD. ***, *p* < 0.001 versus WT sham or *PT*-Sirt6KO sham; ###, *p* < 0.001 versus WT UUO. Sham, sham-operated mice; UUO, unilateral ureteral obstruction; WT, wild-type; *PT*-Sirt6KO, proximal tubule-specific Sirt6 knockout; α-SMA, α-smooth muscle actin; FSP-1, fibroblast-specific protein-1; DAPI, 4′,6-diamidino-2-phenylindole.

**Figure 4 cells-11-01477-f004:**
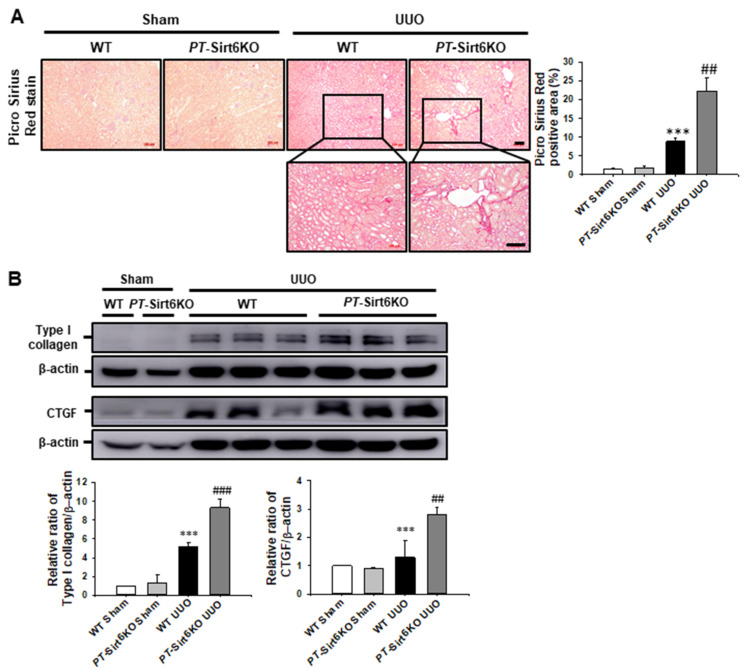
**Proximal tubule-specific loss of Sirt6 increases UUO-induced extracellular matrix deposition.** (**A**) Representative sections of kidneys from sham- and UUO-operated *Sirt6*^flox/flox^; *Ggt1-Cre^-^* (WT) and *Sirt6*^flox/flox^; *Ggt1-Cre^+^* (*PT*-Sirt6KO) mice were stained with Picro Sirius red. Scale bar = 100 μm. The bar graph shows Picro Sirius red-positive areas (%) in the sham and UUO kidneys from ten randomly chosen, non-overlapping fields at a magnification of 100× (n = 15 per group). (**B**) Representative Western blot analysis of type I collagen and CTGF expression in the kidneys from sham- and UUO-operated *Sirt6*^flox/flox^; *Ggt1-Cre^-^* (WT) and *Sirt6*^flox/flox^; *Ggt1-Cre^+^* (*PT*-Sirt6KO) mice. The bar graph shows the densitometric quantification presented as the relative ratio of each protein to β-actin. The relative ratio measured in the kidneys from sham-operated WT mice is arbitrarily presented as 1. Data are expressed as mean ± SD. ***, *p* < 0.001 versus WT sham or *PT*-Sirt6KO sham; ##, *p* < 0.01 and ###, *p* < 0.001 versus WT UUO. Sham, sham-operated mice; UUO, unilateral ureteral obstruction; WT, wild-type; *PT*-Sirt6KO, proximal tubule-specific Sirt6 knockout; CTGF, connective tissue growth factor.

**Figure 5 cells-11-01477-f005:**
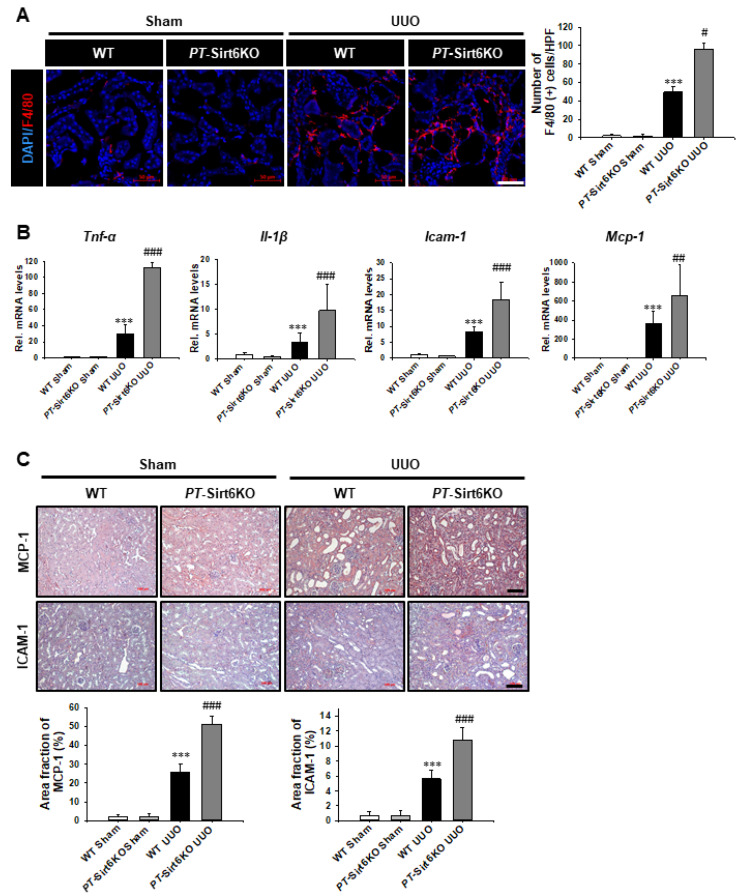
**Proximal tubule-specific loss of Sirt6 aggravates UUO-induced inflammation.** (**A**) Representative sections of kidneys from sham- and UUO-operated *Sirt6*^flox/flox^; *Ggt1-Cre^-^* (WT) and *Sirt6*^flox/flox^; *Ggt1-Cre^+^* (*PT*-Sirt6KO) mice were immunofluorescence stained with F4/80 (red). The nucleus was stained by DAPI (blue). Scale bar = 50 μm. The bar graph shows the number of F4/80 (+) cells in the sham- and UUO-operated kidneys from ten randomly chosen, non-overlapping fields at a magnification of 400× (n = 15 per group). (**B**) Quantitative RT-PCR analyses for *Tnf-**α*, *Il-1β*, *Icam-1,* and *Mcp-1* were performed using mRNA from the sham- and UUO-operated WT and *PT*-Sirt6KO kidneys (n = 15). Data are expressed as mean ± SD. (**C**) Representative sections of kidneys from sham- and UUO-operated *Sirt6*^flox/flox^; *Ggt1-Cre^-^* (WT) and *Sirt6*^flox/flox^; *Ggt1-Cre^+^* (*PT*-Sirt6KO) mice were stained with MCP-1 and ICAM-1. Scale bar = 100 μm. The bar graph shows area fractions (%) of MCP-1 and ICAM-1 in the sham and UUO kidneys from ten randomly chosen, non-overlapping fields at a magnification of 200× (n = 15 per group). ***, *p* < 0.001 versus WT sham or *PT*-Sirt6KO sham; #, *p* < 0.05, ##, *p* < 0.01 and ###, *p* < 0.001 versus WT UUO. Sham, sham-operated mice; UUO, unilateral ureteral obstruction; WT, wild-type; *PT*-Sirt6KO, proximal tubule-specific Sirt6 knockout; *Tnf-**α*, tumor necrosis factor-α; *Il-1β*, interleukin-1β; *Icam-1*, intercellular adhesion molecule-1; *Mcp-1*, monocyte chemoattractant protein-1.

**Figure 6 cells-11-01477-f006:**
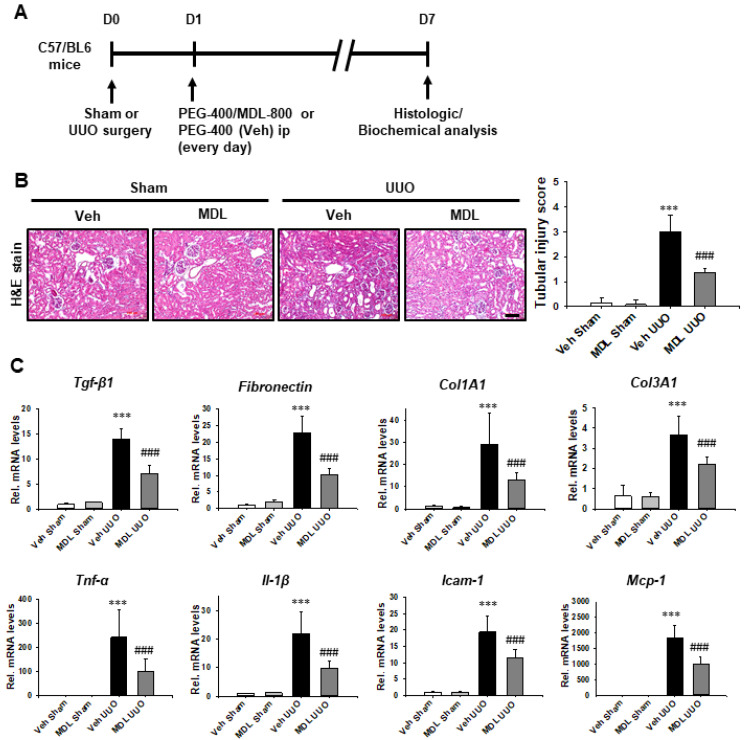
**MDL-800 decreases UUO-induced renal tubulointerstitial inflammation and fibrosis.** (**A**) Scheme for an experimental approach. C57/BL6 mice were divided into four groups (n = 15 per group). One day after UUO surgery, we treated MDL-800 (10 mg/kg) or Veh by daily intraperitoneal injection, a process that continued up to 7 d after UUO surgery. (**B**) Representative sections of kidneys from sham- and UUO-operated C57/BL6 mice treated with Veh or MDL-800 were stained with Hematoxylin and Eosin (H&E). Scale bar = 100 μm. The bar graph shows semi-quantitative scoring of tubular injury by H&E in the sham and UUO-operated kidneys at a magnification of 200×. Ten randomly chosen, non-overlapping fields were quantified (n = 15 per group). Data are expressed as mean ± SD. (**C**) Quantitative RT-PCR analyses for *Tgf-β1, fibronectin, Col1A1, Col3A1, Tnf-**α*, *Il-1β*, *Icam-1,* and *Mcp-1* were performed using mRNA from the sham- and UUO-operated kidneys (n = 15). Data are expressed as mean ± SD. ***, *p* < 0.001 versus Veh sham or MDL sham; ###, *p* < 0.001 versus Veh UUO. Sham, sham-operated mice; UUO, unilateral ureteral obstruction; Veh, vehicle; MDL, MDL-800; *Tnf-**α*, tumor necrosis factor-α; *Il-1β*, interleukin-1β; *Icam-1*, intercellular adhesion molecule-1; *Mcp-1*, monocyte chemoattractant protein-1; *Tgf-β1*, transforming growth factor-β1; *Col1A1*, collagen type I alpha 1 chain; *Col3A1*, collagen type III alpha 1 chain.

**Figure 7 cells-11-01477-f007:**
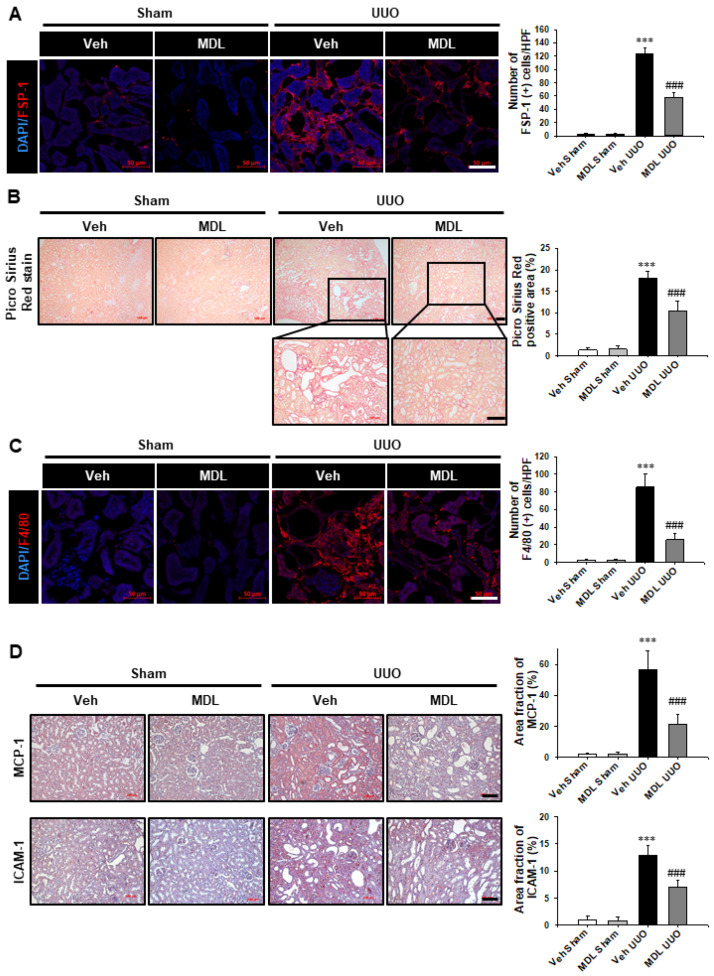
**MDL-800 decreases UUO-induced myofibroblast activation, extracellular matrix deposition, and F4/80 (+) macrophage infiltration.** (**A**) Representative sections of kidneys from sham- and UUO-operated C57/BL6 mice treated with Veh or MDL-800 were immunofluorescence stained with FSP-1 (red). The nucleus was stained by DAPI (blue). Scale bar = 50 μm. The bar graph shows the number of FSP-1 (+) cells in the sham- and UUO-operated kidneys from ten randomly chosen, non-overlapping fields at a magnification of 400× (n = 15 per group). (**B**) Representative sections of kidneys from sham- and UUO-operated C57/BL6 mice treated with Veh or MDL-800 were stained with Picro Sirius red. Scale bar = 100 μm. The bar graph shows Picro Sirius red positive areas (%) in the sham and UUO kidneys from ten randomly chosen, non-overlapping fields at a magnification of 100× (n = 15 per group). (**C**) Representative sections of kidneys from sham- and UUO-operated C57/BL6 mice treated with Veh or MDL-800 were immunofluorescence stained with F4/80 (red). The nucleus was stained by DAPI (blue). Scale bar = 50 μm. The bar graph shows the number of F4/80 (+) cells in the sham- and UUO-operated kidneys from ten randomly chosen, non-overlapping fields at a magnification of 400× (n = 15 per group). Data are expressed as mean ± SD. (**D**) Representative sections of kidneys from sham- and UUO-operated C57/BL6 mice treated with Veh or MDL-800 were stained with MCP-1 and ICAM-1. Scale bar = 100 μm. The bar graph shows area fractions (%) of MCP-1 and ICAM-1 in the sham and UUO kidneys from ten randomly chosen, non-overlapping fields at a magnification of 200× (n = 15 per group). ***, *p* < 0.001 versus Veh sham or MDL sham; ###, *p* < 0.001 versus Veh UUO. Sham, sham-operated mice; UUO, unilateral ureteral obstruction; Veh, vehicle; MDL, MDL-800; FSP-1, fibroblast-specific protein-1; MCP-1, monocyte chemoattractant protein-1; ICAM-1, intercellular adhesion molecule-1.

**Figure 8 cells-11-01477-f008:**
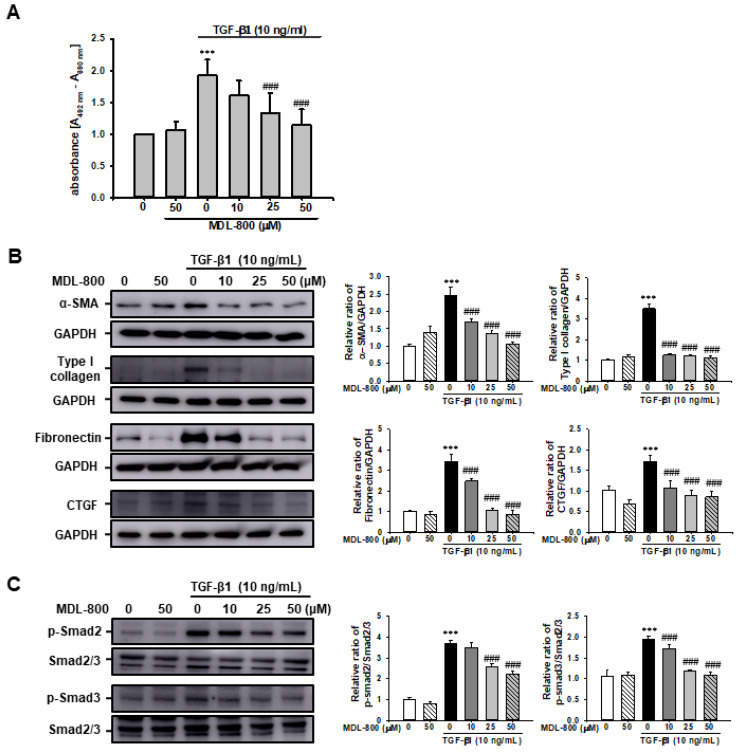
**Effect of MDL-800 on TGF-****β1-induced extracellular matrix protein expression in HK2 cells.** (**A**) HK2 cells were treated with vehicle (Veh) or TGF-β1 (10 ng/mL), with or without MDL-800 at indicated doses (10, 25, or 50 μM). After 24h of treatment, cell proliferation was measured by XTT assay. Data are expressed as mean ± SD for three independent experiments in triplicates. (**B**) Representative Western blot for α-SMA, type I collagen, fibronectin, and connective tissue growth factor (CTGF) from HK-2 cells treated with Veh or TGF-β1 (10 ng/mL), with or without MDL-800 at indicated doses (10, 25, or 50 μM). Treatment with TGF-β1 (10 ng/mL) over 24 h increased the expression of extracellular matrix protein markers. The expression of α-SMA, type I collagen, fibronectin, and CTGF decreased after MDL-800 treatment in a dose-dependent manner. The bar graph shows the densitometric quantification presented as the relative ratio of each protein to GAPDH. Data are presented as mean ± SD. (**C**) Representative Western blot for *p*-Smad2 and p-Smad3 from HK-2 cells treated with Veh or TGF-β1 (10 ng/mL), with or without MDL-800 at indicated doses (10, 25, or 50 μM). Treatment with TGF-β1 (10 ng/mL) over 24 h increased the expression of p-Smad2 and p-Smad3. The expression of p-Smad2 and p-Smad3 decreased in a dose-dependent manner after MDL-800 treatment. The bar graph shows the densitometric quantification presented as the relative ratio of each protein to Smad2/3. Data are presented as mean ± SD. ***, *p* < 0.001 versus Veh or MDL; ###, *p* < 0.001 versus TGF-β1 (10 ng/mL). Veh, vehicle; MDL, MDL-800; TGF-β1, transforming growth factor-β1; α-SMA, α-smooth muscle actin; CTGF, connective tissue growth factor.

**Figure 9 cells-11-01477-f009:**
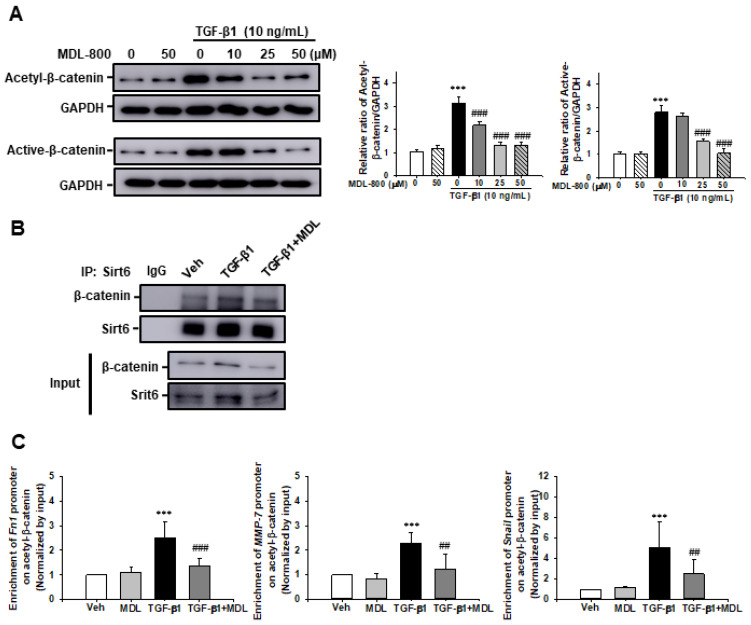
Effect of MDL-800 on TGF-β1-induced β-catenin acetylation and β-catenin target gene promoter enrichment at the site of β-catenin acetylation in HK2 cells. (**A**) Representative Western blot for acetyl-β-catenin and active-β-catenin from HK-2 cells treated with Veh or TGF-β1 (10 ng/mL), with or without MDL-800 at indicated doses (10, 25, or 50 μM). Treatment with TGF-β1 (10 ng/mL) over 24 h increased the expression of acetyl-β-catenin and active-β-catenin. The expression of acetyl-β-catenin and active-β-catenin decreased in a dose-dependent manner after the administration of MDL-800 treatment. The bar graph shows the densitometric quantification presented as the relative ratio of each protein to GAPDH. Data are presented as mean ± SD. (**B**) Representative Western blot analysis of β-catenin and Sirt6 from HK-2 cells after Veh or TGF-β1 (10 ng/mL), with or without MDL-800 (50 μM). HK-2 cell lysates were immunoprecipitated against the Sirt6 antibody, and immunoprecipitants were immunoblotted with β-catenin. (**C**) Chromatin immunoprecipitation (ChIP) assay of β-catenin target genes, such as *Fn1*, *MMP7,* and *Snail* promoter regions in HK2 cells, after 24 h treatment of vehicle (Veh), TGF-β1 (10 ng/mL) with or without MDL-800 (50 μM). Immunoprecipitation was performed using acetyl- β-catenin antibody. Normal mouse IgG was used as a negative control and the PCR product of input chromatin (input 5%) before immunoprecipitation was used as a positive control. Quantitative real-time PCR was performed using specific primers for *Fn1*, *MMP7,* and *Snail* promoter regions. All results were normalized to input levels. Data are expressed as mean ± SD from four independent experiments. ***, *p* < 0.001 versus Veh or MDL; ###, *p* < 0.001 versus TGF-β1 (10 ng/mL); ##, *p* < 0.01 versus TGF-β1 (10 ng/mL). Veh, vehicle; MDL, MDL-800; TGF-β1, transforming growth factor-β1; α-SMA, α-smooth muscle actin; CTGF, connective tissue growth factor; Fn1, fibronectin; MMP-7, matrix metalloproteinase-7.

**Table 1 cells-11-01477-t001:** Primer list of target genes used for qRT-PCR in this study.

Gene	Accession No.	Forward Primer (5′–3′)	Reverse Primer (5′–3′)
*Tnf* *-a*	NM_013693.3	AGGGTCTGGGCCATAGAACT	CCACCACGCTCTTCTGTCTAC
*Il-1* *β*	NM_008361.4	GGTCAAAGGTTTGGAAGCAG	TGTGAAATGCCACCTTTTGA
*Icam-1*	NM_010493	TTTTGGAGCTAGCGGACCAG	CCGCTCAGAAGAACCACCTT
*Mcp-1*	NM_011333	CAGCCAGATGCAGTTAACGC	TTCTTGGGGTCAGCACAGAC
*Tgf-β1*	NM_011577.2	CTGCTGACCCCCACTGATAC	GGGGCTGATCCCGTTGATTT
*Fibronectin*	NM_010233.2	CGAGGTGACAGAGACCACAA	CTGGAGTCAAGCCAGACACA
*Col1A1*	NM_007742.4	GTTTGGAGAGAGCATGACCGA	TGGACATTAGGCGCAGGAA
*Col3A1*	NM_009930.2	AAAGGGGCTGGAAAGTGAGG	AGCACCATCAGTTGTCCCTG

## Data Availability

Not applicable.

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
