# Peer review of "Loss of Proximal Tubular Sirtuin 6 Aggravates Unilateral Ureteral Obstruction-Induced Tubulointerstitial Inflammation and Fibrosis by Regulation of β-Catenin Acetylation"

_cells, 2022, doi:10.3390/cells11091477_

Round 1
Reviewer 1 Report
The data are of high quality and the results rather straightforward. The experiments are well controlled, the statistical treatment appropriate and the conclusions drawn valid.
Author Response
We appreciate your reviewing our manuscript and giving good comments.

Reviewer 2 Report
This is an interesting in vivo study addressing the proximal tubule-specific role of Sirt6. There are, however, weaknesses that are summarized below:
- The authors should discuss when the Cre transgene is expressed. For example, is it expressed during embryogenesis?
- The authors should address/discuss the effect of Sirt6 loss in proximal tubule biology. Does it affect their development and function?
- There is no dispute that the loss od Sirt6 leads to an aggravated phenotype. However, a direct molecular link between Sirt6 loss in the proximal tubules and the severity of the phenotype is missing. For example, the authors should isolate proximal tubule cells from wild-type and knockout mice and compare with qPCR the expression of pro-inflammatory cytokines. The qPCRs shown might simply reflect the increased infiltration of immune cells and could simply be the result rather than he cause of increased inflammation.
- Systemic administration of MDL-800 does not prove that the proximal tubule cells contribute to the severity of the disease. Again, the expression of the indicated genes in section 3.4 should be assessed in proximal tubule cells and not total kidney tissue.
- For the experiments shown in section 3.5, a control cell which is not of proximal tubule origin should be used. This experiment does not prove that the Sirt6 functions specifically in the proximal tubule cells are the cause of aggravated disease in vivo.
- Image quality in all figures us extremely poor.
- The language needs extensive editing.
- The text does not have a universal formatting.
Author Response
Reviewer #2
This is an interesting in vivo study addressing the proximal tubule-specific role of Sirt6. There are, however, weaknesses that are summarized below:
- The authors should discuss when the Cre transgene is expressed. For example, is it expressed during embryogenesis?
Answer) Thank you for your kind comments. The γ-glutamyl transferase Cre mice (Tg-(Ggt1-Cre)M3Egn/J) were initially generated and donated to the Jackson Laboratory by Eric Neilson (J Clin Invest 110(3):341-50). These γGT Cre transgenic mice express Cre transcripts only in the kidney and after postpartum day 14. Therefore, we add this description in the materials and methods section and the discussion.
Page 3, line 81~83: Iwano et al. reported that Cre transcripts are only in the kidney, and after postpartum day 14, of γ-glutamyl transferase Cre transgenic mice express [21].
Page 20~22, line511~517: In the present study, we use γ-glutamyl transferase Cre mice, which express Cre transcripts, including proximal tubules, only in their cortical kidneys, and not in other tissues (e.g., brain, liver, spleen, lung, muscle, or adrenal gland). Additionally, their bone marrow cells do not express γ-glutamyl transferase Cre [21]. In a previous mouse single-cell transcriptomics study, approximately 60% of the kidney cells were found to be proximal tubule cells [38, 39]. Immunofluorescence stain and Western blot analysis of our PT-Sirt6KO mice revealed that they showed significantly less Sirt6 expression in than WT mice.
- The authors should address/discuss the effect of Sirt6 loss in proximal tubule biology. Does it affect their development and function?
Answer) Thank you for your kind comments. Our proximal tubule-specific Sirt6 knock-out mice do not show histologic abnormalities in sham, and UUO operated mice. In addition, Iwano et al. report that γ-glutamyl transferase Cre transgenic mice express Cre transcripts after postpartum day 14. Therefore, γ-glutamyl transferase Cre expression does not affect kidney development and its functions. For more accurate results on the effect of Sirt6 loss in proximal tubule biology, inducible Cre mice such as Tamoxifen-inducible Cre mice are helpful. Unfortunately, we do not have tamoxifen-induced Cre transgenic mice.
Page 8, line 263~264: Histologically, neither WT nor PT-Sirt6KO mice show tubular abnormalities in the sham-operated kidneys.
- There is no dispute that the loss od Sirt6 leads to an aggravated phenotype. However, a direct molecular link between Sirt6 loss in the proximal tubules and the severity of the phenotype is missing. For example, the authors should isolate proximal tubule cells from wild-type and knockout mice and compare with qPCR the expression of proinflammatory cytokines. The qPCRs shown might simply reflect the increased infiltration of immune cells and could simply be the result rather than he cause of increased inflammation.
Answer) We agree reviewer’s comments. For the direct molecular link between Sirt6 loss of proximal tubule and phenotypic severity, we need to isolate proximal tubule cells to address this comment. However, our indirect evidence for regulation of inflammatory cell infiltration, MCP-1, and ICAM-1 expression in PT-Sirt6KO mice might be helpful to understand the role of proximal tubule Sirt6 in kidney fibrosis.
- Systemic administration of MDL-800 does not prove that the proximal tubule cells contribute to the severity of the disease. Again, the expression of the indicated genes in section 3.4 should be assessed in proximal tubule cells and not total kidney tissue.
Answer) Thank you for your constructive comments. We use the MDL-800, Sirt6 activator, to evaluate the systemic therapeutic effect on the kidney fibrosis model. Therefore, we have evaluated the proinflammatory cytokine expression in the whole kidney tissue, not in proximal tubule cells alone.
- For the experiments shown in section 3.5, a control cell which is not of proximal tubule origin should be used. This experiment does not prove that the Sirt6 functions specifically in the proximal tubule cells are the cause of aggravated disease in vivo.
Answer) Thank you for your kind comments. However, our in vitro experiment intend to show the protective effect of MDL-800 on TGFβ-induced extracellular matrix expression and the relationship between Sirt6 and β-catenin. Therefore, we use the HK2 cell, the human proximal tubule cell line, as control cells.
- Image quality in all figures us extremely poor.
Answer) We apologize for the poor quality of the picture. We try to upgrade figure quality.
- The language needs extensive editing.
Answer) Thank you for your kind comments. We have extensive English editing.
- The text does not have a universal formatting.
Answer) We use and follow the MDPI’s journal format.

Reviewer 3 Report
In the manuscript “Loss of proximal tubular Sirtuin 6 aggravates unilateral ureteral obstruction-induced tubulointerstitial 2 inflammation and fibrosis by regulation of β-catenin acetylation” the authors described the protective effect of Sirtuin 6 (Sirt6), specifically in proximal tubule cells, in the development of renal fibrosis. The mechanism involves Sirt6-dependent β-catenin acetylation and TGF-β1/Smad signaling pathway. The role of Sirt6 in pro-fibrotic response has been described in a previous work by other authors [Cai et al., 2020]. Here the authors showed the specific role of Sirt6 present in proximal tubule cells what represents a new relevant contribution.
The work is scientifically sound, relevant for the field of renal pathophysiology and the results are convincing. However, major changes are necessary for acceptance.
Major
1- Despite being enriched in proximal tubule cells, γ-GT is also expressed in other cell types, such as blood mononuclear cells, liver, and spleen. Since overall immune response has been described to be involved in pro-fibrotic response in renal tissue, it is plausible to postulate that Sirt6 depletion in immune cells could also play a role in renal fibrosis in the model used in the present manuscript. We suggest that the authors address this issue in the discussion section.
2- Furthermore, Figure 1A indicate that Sirt6 is downregulated also in LTL-negative tubules, suggesting off-target KO. To be sure that other nephron segments are not involved in the effect of Sirt6 in the pro-fibrotic response we suggest the use of other markers of different nephron segments (NCC for distal tubule and podocin for glomeruli). The limitations of the experimental model used should also be discussed.
3- Please, add a discussion about the differences of Sirtuin expression in the kidney. Is Sirt6 the most expressed? What is the role of Sirt6 in other cell types?
Minor
1- In Figure 9, in the immunoprecipitated samples, the western blot was performed for acetylated β-catenin, or β-catenin? It is described acetylated β-catenin, but the result shows β-catenin. Please check labelling.
2- Careful revision of the written must be performed.
3- Check file format.
4- Images resolution should be checked in the manuscript file
5- We suggest that the abbreviation pSirt6KO could be replaced by PT-Sirt6KO. This could better represent the animal model.
Author Response
Reviewer#3
n the manuscript “Loss of proximal tubular Sirtuin 6 aggravates unilateral ureteral obstruction-induced tubulointerstitial 2 inflammation and fibrosis by regulation of β-catenin acetylation” the authors described the protective effect of Sirtuin 6 (Sirt6), specifically in proximal tubule cells, in the development of renal fibrosis. The mechanism involves Sirt6-dependent β-catenin acetylation and TGF-β1/Smad signaling pathway. The role of Sirt6 in pro-fibrotic response has been described in a previous work by other authors [Cai et al., 2020]. Here the authors showed the specific role of Sirt6 present in proximal tubule cells what represents a new relevant contribution.
The work is scientifically sound, relevant for the field of renal pathophysiology and the results are convincing. However, major changes are necessary for acceptance.
Major
- Despite being enriched in proximal tubule cells, γ-GT is also expressed in other cell types, such as blood mononuclear cells, liver, and spleen. Since overall immune response has been described to be involved in pro-fibrotic response in renal tissue, it is plausible to postulate that Sirt6 depletion in immune cells could also play a role in renal fibrosis in the model used in the present manuscript. We suggest that the authors address this issue in the discussion section.
Answer) Thank you for your kind comments. The γ-glutamyl transferase Cre mice (Tg-(Ggt1-Cre)M3Egn/J) were initially generated and donated to the Jackson Laboratory by Eric Neilson (J Clin Invest 110(3):341-50). These γGT Cre transgenic mice express Cre transcripts only in the kidney and after postpartum day 7~21. This γGT Cre expression occurs in late kidney development, which means that nephrogenesis is almost ended. Therefore, we add this information to the materials and methods section and the discussion.
Page 3, line 81~83: Iwano et al. reported that Cre transcripts are only in the kidney, and after postpartum day 14, of γ-glutamyl transferase Cre transgenic mice express [21].
Page 20~22, line511~517: In the present study, we use γ-glutamyl transferase Cre mice, which express Cre transcripts, including proximal tubules, only in their cortical kidneys, and not in other tissues (e.g., brain, liver, spleen, lung, muscle, or adrenal gland). Additionally, their bone marrow cells do not express γ-glutamyl transferase Cre [21]. In a previous mouse single-cell transcriptomics study, approximately 60% of the kidney cells were found to be proximal tubule cells [38, 39]. Immunofluorescence stain and Western blot analysis of our PT-Sirt6KO mice revealed that they showed significantly less Sirt6 expression in than WT mice.
Page 20, line538-542: One limitation of this study was that we did not use inducible Cre-LoxP systems. However, our γ-glutamyl transferase Cre mice express Cre transcripts after postpartum day 14, which might be the point of complete end of nephrogenesis [21]. In addition, our PT-Sirt6KO mice show grossly and histologically normal findings. If further studies are required to address the role of Sirt6 in kidney disease, we may use a kidney-specific inducible Cre mouse or CRISPR system.
- Furthermore, Figure 1A indicate that Sirt6 is downregulated also in LTL-negative tubules, suggesting off-target KO. To be sure that other nephron segments are not involved in the effect of Sirt6 in the pro-fibrotic response we suggest the use of other markers of different nephron segments (NCC for distal tubule and podocin for glomeruli). The limitations of the experimental model used should also be discussed.
Answer) Thank you for your kind comments. As a previous study by Iwano et al., these γGT Cre transgenic mice express in the cortical kidney, including proximal tubules. So, LTL-negative tubules also might have a negative stain for Sirt6. Our limitation of antibodies availability for NCC or podocin leads us that we cannot properly presentation of Figure 1A. However, our Western blot data suggest PT-Sirt6KO mouse show significantly decreased Sirt6 expression compared to WT mice. We describe our limitation of study in the discussion section.
Page 19, line 509~515: In the present study, we use γ-glutamyl transferase Cre mice, which express Cre transcripts, including proximal tubules, only in their cortical kidneys, and not in other tissues (e.g., brain, liver, spleen, lung, muscle, or adrenal gland). Additionally, their bone marrow cells do not express γ-glutamyl transferase Cre [21]. In a previous mouse single-cell transcriptomics study, approximately 60% of the kidney cells were found to be proximal tubule cells [38, 39]. Immunofluorescence stain and Western blot analysis of our PT-Sirt6KO mice revealed that they showed significantly less Sirt6 expression in than WT mice.
- Please, add a discussion about the differences of Sirtuin expression in the kidney. Is Sirt6 the most expressed? What is the role of Sirt6 in other cell types?
Answer) Thank you for your kind comments. We add the role of sirtuins in the kidneys. And our discussion shows the role of the Sirt6 in different types of cells after kidney injury.
Page 9, line 494~498: Several research reports have focused on the role of sirtuins in the maintenance of renal homeostasis [11]. In the kidney, Sirt1 is the most widely studied, as it is expressed throughout the glomerulus and tubules and maintains structural and functional integrity [11, 31, 32]. Sirt3 is another important mitochondrial sirtuin, responsible for maintaining mitochondrial dynamics, energy homeostasis, and antioxidant defenses in proximal and distal tubules [18, 33].
Minor
- In Figure 9, in the immunoprecipitated samples, the western blot was performed for acetylated β-catenin, or β-catenin? It is described acetylated β-catenin, but the result shows β-catenin. Please check labelling.
Answer) Thank you for your kind comments. We check and correct labelling.
Page 18, line 469~472: (B) Representative Western blot analysis of β-catenin and Sirt6 from HK-2 cells after Veh or TGF-β1 (10 ng/mL), with or without MDL-800 (50 μM). HK-2 cell lysates were immunoprecipitated against the Sirt6 antibody, and immunoprecipitants were immunoblotted with β-catenin.
- Careful revision of the written must be performed.
Answer) Thank you for your kind comments. We have extensive English editing.
- Check file format.
Answer) We use and follow the MDPI’s journal format.
- Images resolution should be checked in the manuscript file.
Answer) We apologize for the poor quality of the picture. We try to upgrade figure quality.
- We suggest that the abbreviation pSirt6KO could be replaced by PT-Sirt6KO. This could better represent the animal model.
Answer) Thank you for your kind comments. We replace the abbreviation pSirt6KO to PT-Sirt6KO to better represent our experimental animal model.

Round 2
Reviewer 2 Report
The authors have tried to address this reviewer's comments. Athough not at 100%, they have largely succeeded.
Reviewer 3 Report
The authors addressed all the questions promptly. The work is relevant and bring new information about the role of proximal tubule Sirt6 on the tubule-interstitial inflammation and fibrosis. Therefore, I recommend its acceptance.